# Neck Circumference for NAFLD Assessment during a 2-Year Nutritional Intervention: The FLiO Study

**DOI:** 10.3390/nu14235160

**Published:** 2022-12-04

**Authors:** Mariana Elorz, Alberto Benito-Boilos, Bertha Araceli Marin, Nuria Pérez Díaz del Campo, Jose Ignacio Herrero, Jose Ignacio Monreal, Josep A. Tur, J. Alfredo Martínez, Maria Angeles Zulet, Itziar Abete

**Affiliations:** 1Department of Radiology, Clínica Universidad de Navarra, 31008 Pamplona, Spain; 2Navarra Institute for Health Research (IdiSNA), 31008 Pamplona, Spain; 3Centre for Nutrition Research, Department of Nutrition, Food Sciences and Physiology, Faculty of Pharmacy and Nutrition, University of Navarra, 31008 Pamplona, Spain; 4Liver Unit, Clínica Universidad de Navarra, 31008 Pamplona, Spain; 5Biomedical Research Centre Network in Hepatic and Digestive Diseases (CIBERehd), 28029 Madrid, Spain; 6Clinical Chemistry Department, Clínica Universidad de Navarra, 31008 Pamplona, Spain; 7Research Group on Community Nutrition and Oxidative Stress, University of Balearic Islands, 07122 Palma, Spain; 8Biomedical Research Centre Network in Physiopathology of Obesity and Nutrition (CIBERobn), Instituto de Salud Carlos III, 28029 Madrid, Spain

**Keywords:** anthropometric measurements, fatty liver disease, nutritional intervention, imaging techniques, long-term follow-up, neck-to-height ratio, non-invasive diagnostic methods, neck-to-weight ratio, FLIO study, steatosis markers

## Abstract

Neck circumference (NC) and its relationship to height (NHtR) and weight (NWtR) appear to be good candidates for the non-invasive management of non-alcoholic fatty liver disease (NAFLD). This study aimed to evaluate the ability of routine variables to assess and manage NAFLD in 98 obese subjects with NAFLD included in a 2-year nutritional intervention program. Different measurements were performed at baseline, 6, 12 and 24 months. The nutritional intervention significantly improved the anthropometric, metabolic and imaging variables. NC was significantly associated with the steatosis degree at baseline (r = 0.29), 6 m (r = 0.22), 12 m (r = 0.25), and 24 m (r = 0.39) (all *p* < 0.05). NC was also significantly associated with visceral adipose tissue at all the study time-points (basal r = 0.78; 6 m r = 0.65; 12 m r = 0.71; 24 m r = 0.77; all *p* < 0.05). NC and neck ratios combined with ALT levels and HOMA-IR showed a good prediction ability for hepatic fat content and hepatic steatosis (at all time-points) in a ROC analysis. The model improved when weight loss was included in the panel (NC-ROC: 0.982 for steatosis degree). NC and ratios combined with ALT and HOMA-IR showed a good prediction ability for hepatic fat during the intervention. Thus, their application in clinical practice could improve the prevention and management of NAFLD.

## 1. Introduction

Non-alcoholic fatty liver disease (NAFLD) is characterized by the accumulation of fatty acids within the hepatocytes as fat vacuoles in subjects consuming little or no alcohol without other causes of liver disease [1]. This entity includes several conditions with ascending severity. The most common condition is simple liver fat accumulation, a non-serious state called fatty liver (simple steatosis). When fat accumulation is associated with liver cell inflammation and different degrees of scarring this is considered a more serious condition called non-alcoholic steatohepatitis (NASH). NASH may lead to severe liver scarring, fibrous bridges might be created (fibrosis) and in more advanced stages regenerative nodules are formed (cirrhosis). Cirrhosis occurs when the liver sustains substantial damage. Subjects at this stage may eventually require a liver transplant [2]. Moreover, hepatic cirrhosis is a potential precursor of hepatocarcinoma. Both steatosis and NASH are reversible and can evolve from one to another. However, when fibrous bridges are generated, the process is irreversible. 

In developed countries, particularly in Europe, the estimated prevalence of NAFLD in the general population is 20–30%, and it increases up to 70% in the case of subjects with obesity or metabolic syndrome [2,3]. Disease progression is slow and asymptomatic; patients are not aware of the presence of the disease until it reaches an irreversible stage when the liver is unable to work properly. 

In the coming years, NASH and alcoholic liver disease will become the most common causes of chronic liver disease all over the world [4]. The gold standard for diagnosing NAFLD and for the assessment of its severity is a liver biopsy. This is an aggressive technique and has possible complications, such as bleeding, which may even endanger the patient’s life. In addition, only a small amount of liver parenchyma is evaluated which may not be representative of the entire liver parenchyma [5,6]. Therefore, non-invasive diagnostic methods are needed, such as radiological techniques, biomarkers, anthropometric measurements or serologic tests that may be used at the population level with low risk and cost, promoting the early detection of the disease [7]. Among the imaging techniques, ultrasound can discriminate between the presence and absence of steatosis, graduating its severity as mild, moderate or severe. It is a technical operator-dependent measurement, but its low cost, availability and non-risk make it an important tool to be considered. Magnetic resonance imaging (MRI) is a technique available in most hospitals and radiology centers. It provides an objective value that is comparable and reproducible. Sensitivity and specificity are high, 96% and 93%, respectively [8], and it can be considered the best imaging technique in the evaluation and quantification of hepatic steatosis [9]. Magnetic resonance imaging is a relatively expensive technique. These techniques are not habitually applied for the prevention or diagnosis of NAFLD which makes an early diagnosis of the disease very difficult. Therefore, new, non-invasive, easy and quick methods applicable in clinical practice and able to predict the disease in the early stages are necessary for the better prevention and management of NAFLD. Several panels of biomarkers and scores have been developed in order to improve the diagnosis of the disease [10,11]. Anthropometric measurements such as neck circumference (NC), neck to height (NHtR) and neck to weight (NWtR) ratios are being analyzed since they might be effective complements to NAFLD screening, favoring an early diagnosis preventing the development and progression of the disease as well as its management during the treatment period. Most of the studies analyzed that have identified surrogate markers of NAFLD are cross-sectional studies. In this sense, the main aim of our research was to determine easy, quick and economic indicators not only for hepatic fat prediction but also for the nutritional management of the disease. In this sense, the objective was to assess the ability of NC and neck ratios to assess liver fat content in participants with NAFLD during a 2-year nutritional intervention program. 

## 2. Materials and Methods

This study is part of a randomized controlled trial registered as FLiO (Fatty Liver in Obesity), (www.clinicaltrials.gov, accessed on 26 October 2022, NCT03183193). It was approved by the Ethics Committee of the Universidad de Navarra, Spain on 24 April 2015 (54/2015) following the Declaration of Helsinki, and the study was conducted following the CONSORT 2010 guidelines. All subjects signed informed consent forms before enrollment in the study.

### 2.1. Study Participants

A total of 98 overweight/obese men and women (age 40–80 years old; BMI ≥ 27.5 kg/m^2^ to <40 kg/m^2^) were enrolled after fulfilling the inclusion criteria of the study (12). All participants underwent an ultrasound examination which confirmed the presence of steatosis and graduated its severity as low, moderate or severe. 

Subjects included in the study were randomized into two different dietary groups following a Mediterranean-style diet to achieve significant weight loss during the 2-year nutritional intervention program. At baseline, the participants were randomly assigned to the American Heart Association (AHA) or the Fatty Liver in Obesity (FLiO) group. A comprehensive assessment was carried out at baseline and at the end of the study. Measurements included anthropometry, body composition by dual-energy X-ray absorptiometry (DXA), biochemical determinations, evaluation of the liver using ultrasonography and magnetic resonance imaging (MRI). Fasting blood samples were properly collected, processed and stored at −80 °C for further analyses. A step-based physical activity recommendation of 10,000 steps/day was given to the participants [12]. Physical activity was estimated using the validated Spanish version of the Minnesota Leisure-Time Physical Activity Questionnaire. The energy expenditure in physical activity was estimated assuming the value of 1 MET (Metabolic Equivalent for Task) = 3.5 mL/kg/min.

### 2.2. Variable Assessment

The determinations of anthropometric measurements (body weight, height, neck and waist circumference), body composition by DXA (Lunar iDXA, encore 14.5, Madison, WI, USA) and blood pressure (Intelli Sense. M6, OMRON Healthcare, Hoofddorp, the Netherlands) were carried out under fasting conditions at the Metabolic Unit of the University of Navarra following standardized procedures. Blood samples were collected, processed and stored at −80 °C for further analyses [13]. Body Mass Index (BMI) was calculated as the body weight divided by the squared height (kg/m^2^). Biochemical determinations, including blood glucose, aspartate aminotransferase (AST), alanine aminotransferase (ALT), gamma-glutamyl transferase (GGT), total cholesterol (TC), high-density lipoprotein cholesterol (HDL-c) and triglyceride (TG) concentrations were measured on an autoanalyzer Pentra C-200 (HORIBA ABX, Madrid, Spain) with specific commercial kits. Insulin was measured using specific ELISA kits (Demeditec; Kiel-Wellsee, Germany) in a Triturus autoanalyzer (Grifols, Barcelona, Spain). Insulin resistance was estimated using the Homeostasis Model Assessment Index (HOMA-IR), which was calculated using the formula elsewhere described [14]. The low-density lipoprotein cholesterol (LDL-c) levels were estimated using the following formula: LDL-c = TC − HDL-c − TG/5. 

### 2.3. Imaging Techniques

The imaging hepatic assessment was performed under fasting conditions by qualified staff at the University of Navarra Clinic. Ultrasonography (Siemens ACUSON S2000 and S3000) was carried out to determine the presence of hepatic steatosis following the previously described methodology [15]. 

Magnetic resonance imaging (Siemens Area 1.5 T, Erlangen Germany) was also used following the Liver Lab protocol to quantify hepatic fat, iron and volume. It consists of a DIXON screening sequence of 3D in-and opposed-phase T2 weighted data acquisition with a two-point Dixon reconstruction. This method offers a visual qualitative assessment of hepatic steatosis. The acquired data allow for a semiquantitative estimation of fat deposition as well as iron overload. Quantitative sequences include multi-echo T2 corrected single breath-hold spectroscopy (HISTO) reproductive values from a single voxel and multi-echo 3D gradient echo (VIBE) imaging with Dixon reconstruction and correction for T2* [2].

### 2.4. Statistical Analyses

The sample size was calculated considering an association between the image techniques and anthropometric variables different from zero. The following formula was used for the sample size calculation: N = [(Zα+Zβ)/C]2 + 3. Thus, considering the probability of making a type I error of 0.05, a probability of making a type II error of 0.20, and hoping to find an association between variables of r = 0.30, a total of 85 subjects were needed to conduct the analysis. 

The normality of the distribution of the evaluated variables was assessed by the Shapiro–Wilk test. The effect of the nutritional intervention and the differences between different study time-points were analyzed using the linear mixed model, an intention-to-treat analysis which prevented any potential bias due to the loss of participants. Pearson or Spearman correlations, according to the variable distribution, were performed to further explore the association between anthropometric variables (neck circumference and neck ratios) and steatosis degree and changes in the hepatic fat at the different study time-points (baseline, 6, 12, 24 months). Receiver operating characteristic (ROC) curves were applied to calculate the power of prediction of a combination panel (neck circumference, ALT and HOMA) for liver fat (by MRI) and liver steatosis (by ultrasonography) at baseline, 6, 12 and 24 months. These results were validated by calculating the optimism-corrected value using Tibshirani’s enhanced bootstrap method described by Harrell [16].

The analyses were carried out using Stata version 12.0 software (StataCorp, College Station, TX, USA). All *p*-values presented are two-tailed and were considered statistically significant at *p* < 0.05.

## 3. Results

A total of 98 overweight/obese participants began the nutritional intervention, 76 reached the 6-month visit, 72 the 12-month visit and 58 completed the nutritional intervention program (Figure 1). 

Both diets improved the anthropometric, biochemical and hepatic variables during the intervention with no relevant differences between the dietary groups, as demonstrated by Marin-Alejandre et al., 2021 [13]. Thus, the data from the dietary groups were combined to promote the statistical power to carry out the aim of the study. The effect of the nutritional intervention program was significant on the anthropometric variables (body weight, BMI, waist circumference) and body composition (total body fat and visceral fat content) (Table 1). Neck circumference (NC) and neck-to-height ratio (NHtR) were significantly decreased after 6 and 12 months of intervention, however, no significant differences were observed after the 24-month follow-up. The neck-to-weight ratio (NWtR) significantly increased during all study time-points (Table 1). The glucose profile was significantly improved (glucose, insulin and HOMA-IR) after the 24-month follow-up program while the lipid profile did not significantly change from baseline values (Table 1). Regarding hepatic status, ALT and GGT significantly decreased during the intervention while the AST value was not modified. Hepatic fat and hepatic volume were significantly improved during the study (Table 1).

To assess the relationship between NC and neck ratios with hepatic steatosis a correlation analysis was performed at all the study time-points. NC and NHtR were significantly associated with the steatosis degree at baseline (r = 0.29; r = 0.32), 6 (r = 0.22; r = 0.39), 12 (r = 0.25; r = 0.46) and 24 months (r = 0.39; r = 0.62), respectively, while NWtR was only associated with the steatosis degree at 12 (r = 0.25) and 24 months (r = 0.26). On the other hand, the slight changes observed in NC and neck ratios during the intervention were significantly associated with the changes observed in hepatic fat content (MRI) at all the study time-points. NC was strongly and significantly associated with the visceral adipose tissue at all the study time-points (basal r = 0.787, *p* < 0.0001; 6 m r = 0.657, *p* < 0.0001; 12 m r = 0.718, *p* < 0.0001; 24 m r = 0.771, *p* < 0.0001). More variables were also analyzed, especially biochemical variables. The most important correlations were observed with insulin, ALT and the HOMA-IR. These variables were significantly associated with the hepatic fat content and visceral adipose tissue at all the study time-points; however, NC maintained the higher association, especially with visceral adipose tissue.

The potential prediction of anthropometric variables (NC, NHtR, NWtR) for hepatic fat content (Table 2) and steatosis degree (Table 3) was assessed by means of a Receiver Operating Curve (ROC) analysis. This analysis was performed at all the study time points. The combination panel, made up of NC or NHtR or NWtR, ALT levels and HOMA-IR, showed a steady, good predictive value for hepatic fat content (Table 2) and steatosis degree (Table 3) at all the study time-points. The predictive ability of these combination panels improved during the nutritional intervention, showing the highest predictive ability for both liver fat content (ROC: 0.85–0.90) and steatosis degree (ROC: 0.95–0.97) at the end of the intervention (Table 2 and Table 3). When the models were adjusted by the weight loss percentage the predictive scores were improved in both cases for the hepatic fat content (Figure 2) and steatosis degree (Figure 3). These results were validated by calculating the optimism-corrected value using Tibshirani’s enhanced bootstrap method described by Harrell. 

## 4. Discussion

This study is a randomized controlled trial that involved 98 patients with ultrasound-proven steatosis. All participants followed two different energy-restricted diets: the AHA and FLIO diets, both with 30% energy restriction. The intervention lasted 24 months with assessment visits at baseline, 6, 12 and 24 months. The most important finding obtained was that anthropometric and biochemical variables such as NC, NHtR or NWtR combined with ALT levels and HOMA-IR resulted in a combination panel able to predict the hepatic fat content and the steatosis degree at all the study time-points. Moreover, this predictive ability was improved when the weight loss achieved during the nutritional intervention was also considered in the models. The utility of NC and neck ratios as a viable and low-cost alternative for the assessment of fat accumulation in the hepatic tissue has also been analyzed by other authors [17,18,19]. In a cross-sectional study including 2761 subjects, NC was significantly wider in NAFLD patients than in subjects with other metabolic conditions or healthy controls [19]. More recently, NC was shown to be significantly associated with central obesity, hypertension, hypertriglyceridemia, impaired fasting glucose and low serum high-density lipoprotein level, as well as metabolic syndrome [11,20,21,22]. A survey conducted on the prevalence of metabolic diseases and risk factors in East China showed that NC was an independent indicator for NAFLD in normal weight men [10]. Another study observed that the NC of individuals with one metabolic syndrome component was lower than those with three or more and cut-off points (39.5 cm for men and 33.3 cm for women) were established for metabolic syndrome prediction [19].

NC has been suggested as an important and simple measurement reflecting the deposition of subcutaneous fat in the neck or fat surrounding the respiratory tract that can help to determine the degree of obesity, particularly upper body adiposity. The upper-body subcutaneous adipose tissue, estimated by NC, is a unique fat deposit that confers additional metabolic risks beyond generalized and abdominal adiposity [18,20,21,22,23,24]. NC, as well as neck ratios, have been strongly associated with insulin levels, HOMA-IR, lipid alterations and diabetes [17,21,25,26,27]. In the Framingham Heart Study, participants with a large NC had various cardiometabolic risk factors when compared to those with a small NC, even after adjustment for visceral adipose tissue and BMI. The Korean Genome and Epidemiologic Study observed that NC was associated with type 2 diabetes incidence. Participants in the highest NC quartile showed the highest diabetes incidence in comparison with participants from the other quartiles [28]. In the present work, the HOMA-IR was an important variable in the predictive model for liver fat and liver steatosis at all the study time-points. The combination of HOMA-IR with ALT levels and NC or neck ratios improved the predictive ability more than that observed with each variable separately. Insulin resistance is a known risk factor for NAFLD and the addition of the HOMA-IR to the model confirmed its important role in liver fat accumulation [29,30]. Visceral adipose tissue is considered one of the main risk factors for insulin resistance. In the present study, NC was strongly associated with the visceral adipose tissue at all the study time-points which corroborates its association with metabolic alterations such as insulin resistance.

Lifestyle modification is established as the first-line treatment for NAFLD by scientific societies for the study of liver diseases. A healthy dietary intervention is essential to induce progressive weight loss, reduce liver fat accumulation and improve insulin resistance as well as the associated metabolic comorbidities. The nutritional intervention program applied in the present work was based on an energy restriction of 30% of the total energy requirements and a high adherence to the Mediterranean diet. Data showed a relevant weight loss, especially after the first 6 months of intervention, which induced at the same time important liver fat reductions, transaminases modification and improvements in the glucose profile. Except for NWtR, NC and NHtR were not significantly modified during the intervention; however, both measures were associated with the steatosis degree and the slight changes observed in these variables (NC, NHtR and NWtR) were strongly associated with the change observed in hepatic fat content (MRI) at all the study time-points suggesting that NC and neck ratios seem to be sensitive indicators of hepatic fat accumulation and could be used for the assessment of NAFLD during a nutritional intervention. 

Receiver operating characteristic curve analysis showed that the combination of neck measurements with ALT and HOMA-IR was a good predictive panel of the steatosis degree and liver fat content. These predictions were even improved after the 24-month nutritional intervention program when weight loss was also added to the models. The areas under the ROC curves were between 0.90 and 0.91 for the liver fat content and 0.97 and 0.98 for the steatosis degree. Neck circumference and neck ratios, in addition to being important indicators of hepatic fat content, could also be considered good markers for the monitoring of NAFLD subjects that are included in a nutritional intervention program. 

The main underlying mechanisms suggested are that upper body obesity causes metabolic abnormalities, including increased circulating free fatty acids (FFAs). The excess FFAs may contribute to the development of fatty liver disease by contributing to triglyceride formation and storage in the liver (24). It has been shown that 59% of hepatic fat is derived from circulating FFAs, with lesser contributions from de novo lipogenesis (26%) and diet (15%). In addition, excess FFAs may induce insulin resistance, which is thought to be related to the first “hit” in the multistep pathogenesis of NAFLD, and by increasing oxidative stress, thereby trigger the inflammatory response and progressive liver damage (18). More studies indicate that NC is closely correlated with glucolipid dysregulation, hyperinsulinemia, HOMA-IR and other CVD risk factors (19). Therefore, the combination of NC or neck ratios with HOMA-IR and ALT levels achieved a higher predictive ability for hepatic status than the variables independently. The confirmation of the utility of these variables for the assessment of NAFLD will make the management of the disease easier in nutritional intervention studies as well as in clinical practice. 

Some limitations of the present study are that liver status was evaluated using only noninvasive techniques instead of liver biopsy; the presence of hepatic steatosis was determined by ultrasonography. Hepatic fat was quantified by magnetic resonance imaging (MRI), and the biopsy procedure was not performed. The proportion of patients lost to follow-up was high, thus, the sample size was considerably reduced especially after 24 months. Thus, an intention-to-treat analysis was applied in order to avoid possible bias due to the missing values. On the other hand, some strengths can be mentioned. Participants were carefully selected following exclusion and inclusion criteria to avoid a heterogeneous sample. Liver disease was assessed by qualitative (ultrasonography) and quantitative (MRI) methodology in order to achieve a good liver health characterization. Furthermore, there have not been many long-term nutritional intervention studies (2 years) conducted in subjects with NAFLD. 

## 5. Conclusions

The present study shows that NC and neck ratios are easy anthropometric measurements that, in combination with routine biochemical variables (ALT and HOMA-IR), showed good prediction ability of the hepatic fat content. More longitudinal research studies should be performed to confirm the validity and sensitivity of these variables since their applicability in clinical practice would improve the management of NAFLD. 

## Figures and Tables

**Figure 1 nutrients-14-05160-f001:**
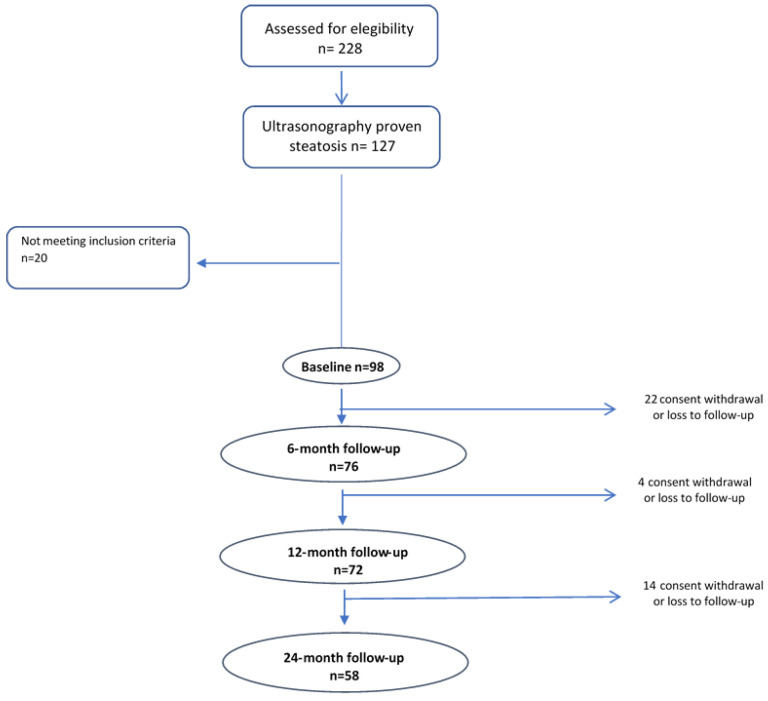
Flowchart of the nutritional intervention.

**Figure 2 nutrients-14-05160-f002:**
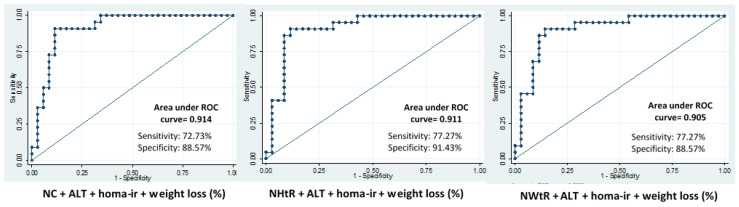
Receiver operating characteristic curve (ROC) analysis considering the hepatic fat content (MRI-histo) as the binary dependent variable, and neck and neck ratios combined with ALT, HOMA-IR and total weight loss (%) as independent variables after 24 months of follow-up.

**Figure 3 nutrients-14-05160-f003:**
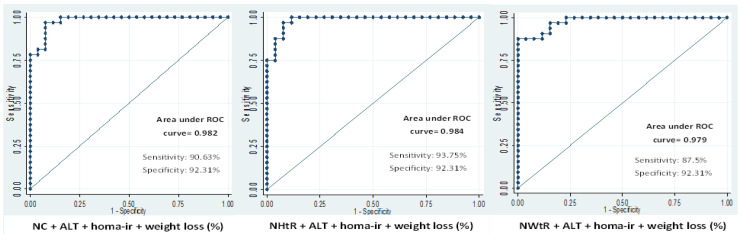
Receiver operating characteristic curve (ROC) analysis considering the steatosis degree as the binary dependent variable (0 = grade 1; 1 = grade 2 + 3) and neck and neck ratios combined with ALT, HOMA-IR and total weight loss (%) as independent variables after 24 months of follow-up.

**Table 1 nutrients-14-05160-t001:** Descriptive variables (anthropometric, body composition, biochemical and imaging technique variables) of study participants at baseline and after 6, 12 and 24 months of nutritional intervention.

	Study Time-Points				
Variables	Basal (n = 98)	6 m (n = 76)	12 m (n = 72)	24 m (n = 58)	*p*-Mixed Model
Anthropometric variables					
Weight (kg)	94.9 ± 13.9	85.4 ± 13.1 *	86.8 ± 14.2 *	89.4 ± 14.8 *	<0.0001
BMI (kg/m^2^)	33.4 ± 3.7	30.1 ± 3.8 *	30.7 ± 4.3 *	31.5 ± 4.8 *	<0.0001
Waist circ (cm)	109.1 ± 8.8	99.7 ± 9.7 *	96.9 ± 19.2 *	105.0 ± 11.9 *	0.001
Total body fat (kg)	38.4 ± 8.6	31.1 ± 9.0 *	32.2 ± 9.4 *	34.6 ± 10.1 *	<0.0001
Visceral fat (kg)	2.2 ± 0.9	1.5 ± 0.7 *	1.7 ± 1.0 *	1.9 ± 1.0 *	<0.0001
Neck circ (cm)	39.6 ± 3.7	38.0 ± 3.5 *	38.2 ± 3.5 *	39.4 ± 4.0	0.172
NHtR	23.4 ± 1.8	22.6 ± 1.8 *	22.8 ± 1.8 *	23.4 ± 2.1	0.216
NWtR	0.41 ± 0.04	0.45 ± 0.05 *	0.44 ± 0.04 *	0.44 ± 0.05 *	<0.0001
Biochemical variables					
Glucose (mg/dL)	103.2 ± 17.1	93.8 ± 12.6 *	94.2 ± 17.6 *	96.2 ± 19.4 *	<0.0001
Insulin (mU/L)	17.3 ± 8.2	11.2 ± 7.2 *	12.5 ± 7.2 *	12.0 ± 0.9 *	<0.0001
HOMA-IR	4.5 ± 2.4	2.6 ± 2.0 *	3.1 ± 2.5 *	3.0 ± 2.0 *	<0.0001
Total cholesterol (mg/dL)	191.1 ± 36.5	180.9 ± 41.9 *	180.0 ± 34.1 *	188.6 ± 41.7	0.299
Triglycerides (mg/dL)	129.8 ± 61.1	94.5 ± 50.6 *	105.8 ± 46.9 *	125.9 ± 79.0	0.240
HDL-c (mg/dL)	51.8 ± 13.0	53.8 ± 12.8 *	54.8 ± 13.2 *	53.5 ± 13.6	0.237
LDL-c (mg/dL)	113.2 ± 32.2	107.7 ± 36.0	104.2 ± 29.4 *	109.9 ± 32.5	0.353
Hepatic variables					
ALT (IU/L)	33.2 ± 17.1	22.2 ± 8.8 *	25.0 ± 12.0 *	26.9 ± 15.1 *	0.001
AST (IU/L)	25.3 ± 10.1	21.7 ± 7.3 *	22.9 ± 8.7	24.4 ± 7.7	0.577
GGT (IU/L)	38.6 ± 28.6	27.3 ± 34.3 *	28.4 ± 19.6 *	29.4 ± 31.3	0.025
Hepatic fat (hist) (%)	10.5 ± 6.3	5.8 ± 4.0 *	6.7 ± 5.7 *	7.5 ± 6.1 *	<0.0001
Hepatic fat (dix) (%)	7.8 ± 8.2	3.2 ± 3.2 *	5.3 ± 4.8 *	5.7 ± 4.5	0.043
Hepatic volumen (cm^3^)	1757.6 ± 399.9	1591.2 ± 318.5 *	1620.2 ± 380.3 *	1660.1 ± 493.4 *	<0.0001

A linear mixed model was used to assess the effect of the intervention as well as differences between the study time-points. Variables with normal distribution: waist and neck circumference, NHtR and NWtR, total cholesterol and LDL-c. * indicates statistical differences between basal vs 6, 12 and 24 months. NHtR: neck circumference to height ratio; NWtR: neck circumference to weight ratio.

**Table 2 nutrients-14-05160-t002:** Receiver operating characteristic curve (ROC) analysis considering the hepatic fat content as the binary dependent variable, and neck and neck ratios combined with ALT and HOMA as independent variables at baseline and all the study time-points (6, 12 and 24 months of follow-up).

	Hepatic Fat Content (MRI-Dixon)				Hepatic Fat Content (MRI-Histo)			
Combination Panels	Time-Point	Lroc	Sensitivity	Specificity	Time-Point	Lroc	Sensitivity	Specificity
NC + ALT + HOMA-IR	Baseline	0.79	63.6	74.5	Baseline	0.79	85	70.5
	6 m	0.79	28.5	98.4	6 m	0.83	42.1	94.2
	12 m	0.75	38.8	95.9	12 m	0.79	47.3	95.9
	24 m	0.85	56.2	95.0	24 m	0.89	68.1	91.4
NHtR + ALT + HOMA-IR	Baseline	0.81	70.4	78.4	Baseline	0.81	83.3	61.7
	6 m	0.82	28.5	98.3	6 m	0.87	52.6	94.2
	12 m	0.78	61.1	95.9	12 m	0.79	47.3	95.9
	24 m	0.88	56.2	95.0	24 m	0.90	68.1	88.5
NWtR + ALT + HOMA-IR	Baseline	0.79	95.9	80.3	Baseline	0.80	83.3	58.8
	6 m	0.79	28.5	100	6 m	0.81	47.3	96.1
	12 m	0.77	38.8	95.9	12m	0.81	42.1	95.9
	24 m	0.84	50.0	92.5	24 m	0.88	59.1	88.5

NHtR: neck circumference to height ratio; NWtR: neck circumference to weight ratio.

**Table 3 nutrients-14-05160-t003:** Receiver operating characteristic curve (ROC) analysis considering the steatosis degree as the dependent variable (considering “0” = steatosis grade 1 and “1” = steatosis grades 2 and 3), and neck and neck ratios (NHtR and NWtR) combined with ALT and HOMA-IR as independent variables at baseline and all the study time-points (6, 12 and 24 months of follow-up).

Combination Panels	Steatosis Degree			
	Time-Point	Lroc	Sensitivity	Specificity
NC + ALT + HOMA-IR	Baseline	0.78	59.4	83.0
	6 m	0.70	83.3	29.1
	12 m	0.74	79.0	57.1
	24 m	0.95	84.3	84.6
NHtR + ALT + HOMA-IR	Baseline	0.78	59.4	83.0
	6 m	0.73	85.7	50.0
	12 m	0.77	76.7	53.5
	24 m	0.97	93.7	92.3
NWtR + ALT + HOMA-IR	Baseline	0.76	45.9	86.4
	6 m	0.71	80.9	29.1
	12 m	0.75	76.7	57.1
	24 m	0.95	81.2	84.6

NC: neck circumference; NHtR: neck circumference to height ratio; NWtR: neck circumference to weight ratio.

## Data Availability

The data presented in this study are available on request from the corresponding author. The data are not publicly available due to ethical reasons. All the data belong to a private collection named “The FLiO Study”.

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
