# Peer review of "Neck Circumference for NAFLD Assessment during a 2-Year Nutritional Intervention: The FLiO Study"

_nutrients, 2022, doi:10.3390/nu14235160_

Round 1
Reviewer 1 Report
The manuscript does not fall under the category of "Nutritional Methodology and Assessment"
See attached comments.

Author Response
Reviewer 1
This manuscript describes a potential approach to diagnose/predict NAFLD in human based studies. It is proposed that the neck circumference (NC) in combination with other biochemical parameters could serve as surrogate markers of NAFLD. The present manuscript is part of an intervention trial which was conducted to determine if two dietary approaches involving 30% reduction in energy intake will affect NAFLD. Authors have published two manuscripts previously (nutrients 11(10)2543 2019 and Liver International, 7 1532, 2021) from the intervention study showing a benefit of energy reduction on the severity of NAFLD. Extensive methodologies were used to assess liver steatosis including non-invasive and invasive techniques, anthropometric measurements, and biochemical assessments.
In the present manuscript the information was assessed after 6, 12 and 24 months of dietary interventions. The focus appeared to be on the importance of NC and NC to height (NHtR) and NC to weight ratios (NWtR). Various other measurements were taken as described in the previous studies.
Body weight reduced at all-time points however not the NC.
Authors carried out to determine if any combinations of biochemical and NHtR or NWtR will be a sensitive and specific surrogate end point. In the reviewer’s opinion, not seeing changes in NC due to intervention it is not something one can rely on. Why even consider NC?
Authors have used other assessment which measures body fat, liver volume, body weight, insulin and glucose level they are all show favourable effect of energy restriction and appeared to have high predictive values.
I am unclear as to whether authors conducted sensitivity and specificity analyses on other routine biochemical measurements for detecting NAFLD severity?
I understand that if NC had changed due to dietary intervention along with a favourable effect on NAFLD the authors could have proposed NC as a non-invasive and most economical approach to assess NAFLD but it was not the case.
We agree with the reviewer´s comment, the variable neck circumference (NC) did not change significantly at the end of the intervention. It decreased significantly after 6 and 12 months of intervention and almost got back to the initial values after 24 months. Interestingly, NC was correlated with the hepatic fat content and the slight changes observed in this variable were very good associated with the changes observed in the hepatic fat content at all the study time-points. Moreover, NC was strongly and significantly associated with the visceral adipose tissue at all the study time-points (basal r=0.787, p<0.0001; 6 months r=0.657, p<0.0001; 12 months p=0.718, p<0.0001; 24 months p=0.771, p<0.0001). More variables were also analysed, especially biochemical variables. Most important correlations were observed with insulin, ALT and the HOMA-IR. These variables were significantly associated with the hepatic fat content and visceral adipose tissue at all the study-time points; however, NC maintained the higher association especially with visceral adipose tissue. Because of that all these variables have been also considered and included in the ROC analyses, maintaining NC and neck ratios as main predictors. As the reviewer suggests, we also tried to find a simple and easy measurement, with a low evaluation cost able to assess the presence and management of NAFLD during the intervention. In this sense, NC was an easy and low cost anthropometric measurement good correlated with visceral and hepatic fat content as well as with the changes of hepatic fat during the nutritional program. Likewise more authors have described NC as a good indicator of the subcutaneous fat in the upper body as well as its association with cardiometabolic risks (Zanuncio et al., 2021), however, nobody have analysed the role of NC for de assessment of NAFLD during a nutritional intervention.
- Zanuncio VV, Sediyama CMNO, Dias MM, Nascimento GM, Pessoa MC, Pereira PF, Silva MRI, Segheto KJ, Longo GZ. Neck circumference and the burden of metabolic syndrome disease: a population-based sample. J Public Health (Oxf). 2021:fdab197.
There is very little reference to nutrition. The method assessment described in the manuscript may not fall under the umbrella of “Nutritional Methodology and Assessment”.
Sorry for the mistake, the manuscript does not correspond to the “Nutritional Methodology and Assessment” area.
The title of the manuscript seemed to be unjustified.
According to the reviewer´s suggestion we have changed the title of the manuscript.
“Neck circumference for NAFLD assessment during a 2-year nutritional intervention: The FLiO study”
Reviewer 2 Report
The manuscript entitled „Neck circumference in combination with biochemical variables as a surrogate marker of NAFLD: The FLiO study” presents interesting issue, but some problems should be corrected.
Abstract:
Sample size should be specified.
Specific numeric results accompanied by a results of the statistical analysis (p-Value) should be presented.
Introduction:
Line 32 – “Background” word should be removed
Lines 70-71 – “can be good indicators of the hepatic status” – this statement should be specified – against which factor was it validated and what results were obtained
Lines 84-89 – should be removed from the Introduction Section
Materials and methods:
Only for parametric data Authors should present mean and SD, while for non-parametric they should present median, min and max values.
Results:
Only for parametric data Authors should present mean and SD, while for non-parametric they should present median, min and max values.
Tables should be clearly described in footnotes (which distributions are parametric and which are not, which tests were used, etc.)
Discussion:
Authors should in their discussion include 3 areas: (1) compare gathered data with the results by other authors, (2) formulate implications of the results of their study and studies by other authors, (3) formulate the future areas which should be studied.
Limitations of the study should be presented.
Authors’ contributions:
This section must correspond authorship – e.g. there is no such author as A.B.-B. and J.I.M. among Authors of the study. At the same time some Authors of the study are not included into Authors’ contributions.
It seems that contribution of some authors was only minor and they did not participate in preparing manuscript. There is a serious risk of a guest authorship procedure which is forbidden. In such case they should be rather presented in Acknowledgements Section and not be indicated as authors of the study.
Author Response
Reviewer 2:
Comments and Suggestions for Authors:
The manuscript entitled “Neck circumference in combination with biochemical variables as a surrogate marker of NAFLD: The FLiO study” presents interesting issue, but some problems should be corrected.
Abstract:
Sample size should be specified.
The sample size has been included in the abstract.
Specific numeric results accompanied by a result of the statistical analysis (p-Value) should be presented.
According to the reviewer we have included some numeric results of the statistical analysis in the abstract.
Introduction:
Line 32 – “Background” word should be removed.
The word “background has been removed.
Lines 70-71 – “can be good indicators of the hepatic status” – this statement should be specified – against which factor was it validated and what results were obtained.
We have changed the expression in order to clarify this statement.
Lines 84-89 – should be removed from the Introduction Section
Lines have been removed from the introduction section
Materials and methods:
Only for parametric data Authors should present mean and SD, while for non-parametric they should present median, min and max values.
We agree with the reviewer´s comment, however the only table where this could be applied is Table 1. The statistical analysis applied to build up Table 1 was a multilevel approach an intention to treat analysis which is strongly recommended in longitudinal studies with large number of repeated measures. There is not a multilevel approach for non-normal variables, thus intention-to treat analysis should be applied in this type of studies in spite of non-normal distributed variables.
Results:
Only for parametric data Authors should present mean and SD, while for non-parametric they should present median, min and max values.
Tables should be clearly described in footnotes (which distributions are parametric and which are not, which tests were used, etc.)
The statistical method applied to assess the nutritional intervention effect as well as differences between study time-points was an intention to treat analysis (Lineal mixed model). Missing data and large number of repeated measures represent a challenge in longitudinal studies, because of that intention to treat analyses are recommended for this type of studies in spite of most of the variables use to follow a non-normal distribution (Rosato et al., 2021). According to the reviewer we have included in the footnote of the Table 1 which variables are normal distributed, however, we have maintained the statistical approach since there are not statistical multilevel tests for non-normal variables. We have included a table with the median, maximum and minimum of the non-normal variables.
- Rosato R, Pagano E, Testa S, Zola P, di Cuonzo D. Missing data in longitudinal studies: Comparison of multiple imputation methods in a real clinical setting. J Eval Clin Pract 2021;27(1):34-41.
These variables showed a non-normal distribution. Data show the median (min; max)
|
Variables |
Basal (n=98) |
6m (n=76) |
12m (n=72) |
24m (n=58) |
|
Anthropometric variables |
|
|
|
|
|
Weight (kg) |
94.4 (68.8; 134) |
83.6 (64.9; 124.9)* |
84.5 (67.7; 133.2)* |
86.9 (66.9; 129.6)* |
|
BMI (kg/m2) |
32.8 (27.4; 43.7) |
28.9 (23.6; 40.8)* |
29.6 (23.4; 43.5)* |
30.5 (23.3; 42.3)* |
|
Total body fat (kg) |
37.7 (25.3; 65.7) |
30.4 (12.3; 57.8)* |
30.2 (11.4; 63.3)* |
32.8 (11.2; 61.3)* |
|
Visceral fat (kg) |
2.1 (0.5; 5.7) |
1.4 (0.3; 4.4)* |
1.5 (0.3; 5.8)* |
1.7 (0.2; 5.8)* |
|
Biochemical variables |
|
|
|
|
|
Glucose (mg/dl) |
100 (83; 174) |
92 (73; 143)* |
90.5 (64; 185)* |
92 (64; 164)* |
|
Insulin (mU/L) |
16.2 (2.6; 42.2) |
8.6 (2.3; 36.3)* |
10.7 (2.5; 42.4)* |
10.6 (4.2; 42.2)* |
|
HOMA-IR |
4.0 (0.6; 12.2) |
1.9 (0.4; 11.3)* |
2.3 (0.5; 16.0)* |
2.4 (0.7; 10.6)* |
|
Triglycerides (mg/dl) |
122.5 (50; 345) |
83 (43; 341)* |
91 (35; 269)* |
99.5 (35; 418) |
|
HDL-c (mg/dl) |
51.5 (24; 95) |
52 (31; 90)* |
54 (27; 96)* |
52.1 (29; 99.7) |
|
Hepatic variables |
|
|
|
|
|
ALT (IU/L) |
29.5 (10; 106) |
21 (10; 50)* |
21 (4; 64)* |
23.5 (11; 81)* |
|
AST (IU/L) |
24 (11; 77) |
20 (6; 53)* |
21 8 (10; 64) |
23 (13; 51) |
|
GGT (IU/L) |
29 (8; 137) |
18 (6; 271)* |
21 (7; 101)* |
21 (2; 183) |
|
Hepatic fat (hist) (%) |
9.1 (2.5; 30.2) |
4.1 (2.2; 19.6)* |
4.2 (2.3; 32)* |
5.7 (2; 32.4)* |
|
Hepatic fat (dix) (%) |
5.7 (0.2; 62.5) |
5.7 (0.5; 16)* |
3.3 (0.4; 26.1)* |
3.9 (1.8; 22.7) |
|
Hepatic volumen (cm3) |
1701 (1015; 2897) |
1523 (988; 2669)* |
1557 (1066; 3121)* |
1572 (1063; 3501)* |
Discussion:
Authors should in their discussion include 3 areas: (1) compare gathered data with the results by other authors, (2) formulate implications of the results of their study and studies by other authors, (3) formulate the future areas which should be studied.
Limitations of the study should be presented.
We have improved the discussion section considering the reviewer´s comment. Likewise a paragraph describing some of the limitations of the study has been included.
Authors’ contributions:
This section must correspond authorship – e.g. there is no such author as A.B.-B. and J.I.M. among Authors of the study. At the same time some Authors of the study are not included into Authors’ contributions.
It seems that contribution of some authors was only minor and they did not participate in preparing manuscript. There is a serious risk of a guest authorship procedure which is forbidden. In such case they should be rather presented in Acknowledgements Section and not be indicated as authors of the study.
The authors´ contribution has been corrected. The name of some authors was incomplete in the authors´ section and then the initials in the authors´ contribution did not fit. Now the authors and authors´ contributions are correct.
